# The Role of the Piezo1 Mechanosensitive Channel in the Musculoskeletal System

**DOI:** 10.3390/ijms24076513

**Published:** 2023-03-30

**Authors:** Beatrix Dienes, Tamás Bazsó, László Szabó, László Csernoch

**Affiliations:** 1Department of Physiology, Faculty of Medicine, University of Debrecen, H-4032 Debrecen, Hungary; 2Department of Orthopedics, Faculty of Medicine, University of Debrecen, H-4032 Debrecen, Hungary; 3ELKH-DE Cell Physiology Research Group, H-4032 Debrecen, Hungary

**Keywords:** mechanosensitive channels, Piezo1, Yoda1, calcium homeostasis, skeletal muscle, cartilage, bone, tendon, ligament, joint

## Abstract

Since the recent discovery of the mechanosensitive Piezo1 channels, many studies have addressed the role of the channel in various physiological or even pathological processes of different organs. Although the number of studies on their effects on the musculoskeletal system is constantly increasing, we are still far from a precise understanding. In this review, the knowledge available so far regarding the musculoskeletal system is summarized, reviewing the results achieved in the field of skeletal muscles, bones, joints and cartilage, tendons and ligaments, as well as intervertebral discs.

## 1. Introduction

The discovery of Piezo channels just over a decade ago [1] not only resulted in the Nobel Prize in Physiology or Medicine in 2021 but also opened a new way to understand mechanotransduction, thus multiplying the scientific interest in the topic. Therefore, articles discussing the role of Piezo channels in different cell types, tissues, and organs both under physiological and pathological conditions follow one another. A significant amount of scientific data is available about the fundamental role of the Piezo1 channel in the development and function of the cardiovascular system [2,3,4,5,6], in the functioning of the pancreas [7] and the excretory organ system [5], in the homeostasis of epithelial cells [8,9,10,11], in innate immunity [12], as well as in the responses of blood cells to mechanical stimuli [13]. Although the number of studies related to the locomotor system has increased significantly in recent years, the role of Piezo1 channels in these organs is still far from being precisely clarified. In this review, after a brief general description of mechanosensitive channels, observations related to Piezo1 in the musculoskeletal system are summarized (Figure 1).

## 2. Mechanosensation—Mechanosensitive and Mechanically Activated Channels

The conversion of mechanical stimuli from the external or internal environment of cells into biological responses, i.e., mechanotransduction, is of fundamental importance both during tissue development and in the physiological processes of mature individuals. Some well-known basic sensing processes, such as hearing, touch, and blood pressure sensing, are now clearly linked to mechanotransducers, but our understanding of mechanosensation is now much broader. In many cell types, the role of mechanosensors has been described in a broad range of cellular functions such as cell migration, proliferation, differentiation, as well as the regulation of vascular development and vascular endothelial cells. Their role in the regulation of pulmonary circulation, in the progression of certain types of cancer, in urinary tract filtration and osmoregulation processes, and in monocyte activation, to name just a few processes without claiming to be exhaustive, has been revealed, although several processes remain to be discovered. The molecular background of this signal transduction has long been of intense interest. Although there are integral membrane proteins, such as GPCRs or membrane-associated phospholipases, that can also be modulated by mechanical stimulus [14,15,16], the main mechanosensors that mediate rapid electrochemical responses to mechanical stimuli are ion channels in several cellular processes [17]. The low expression of these channels, their structural diversity, and the lack of evolutionary conservation hindered the research of mechanosensitive channels in mammals. The mechanosensitive processes discovered in different organisms—from bacteria through fungi to humans—were slowly organized into a more transparent picture, and, despite the diversity of identified ion channels, they were more or less integrated into a unified system that includes several well-defined ion channel families [4,18,19,20].

### Mechanosensitive Ion Channels

Mechanosensitive—or more precisely, mechanically activated—ion channels are manifold in terms of both their structure and function. The only structural feature they have in common is that they contain at least two transmembrane domains. In response to mechanical stimuli exerted on cell membranes, mechanically activated ion channels change their conformations and open in a pore-like manner. According to the “force through lipid” model, changes in membrane tension are recognized [21], while the “force through tether” model requires the contribution of associated proteins of the extracellular matrix and/or cytoskeleton (spring-like tethers) transmitting the force applied on the cell membrane [22,23].

Discovery of mechanosensitive channels started with identification of bacterial MscL (mechanosensitive channel large conductance) and MscS (mechanosensitive channel small conductance) [19,24]. These non-selective channels demonstrate the biophysics of mechanosensitive channels but are not expressed in mammals. Mammalian mechanosensitive ion channels were first described in tissue-cultured embryonic chick skeletal muscle using the patch-clamp technique [25].

Since then, disparate mechanosensitive ion channel superfamilies have been characterized in mammals, including the two pore-domain potassium ion channels (K2P/KCNK), whose three members are inherently mechanosensitive (TREK1, TREK2, and TRAAK) [26]. Some members of the transient receptor potential (TRP) non-selective cation channels are also suggested to have mechanosensitive properties [17], the most obvious evidence being for TRPV4 [27]. The epithelial sodium channel (ENaC) [28] and transmembrane protein 63 (TMEM63) Ca^2+^-activated chloride channels [29,30] are also known as mechanosensitive. The discovery of mechanically activated Piezo channels (e.g., Piezo1 and Piezo2) which are inherently mechanosensitve [31] has significantly contributed to the interpretation of mechanotransduction mechanism. Here, we mainly focus on the role of Piezo1.

For comprehensive and detailed summaries on mechanosensitive channels, we direct the readers’ attention to, e.g., the works of J.M. Kefauver et al. [32] and Ranade et al. [18].

It is, however, worth mentioning that in addition to mechanically activated ion channels, many other channels, the gating of which primarily depends on non-mechanical effects (ligand binding, voltage change), have also been shown to be influenced by mechanical effects and are, therefore, mechanosensitive. Their gating domain, however, is located in the membrane, so its tension modifies the opening of the channel. Examples include the following channels: Kv1.1 [33], Cav [34], Nav1.5 [35,36], HCN [37]. These channels are not considered inherently mechanically activated channels.

## 3. Piezo Channels in the Animal Kingdom

Living organisms constantly face environmental mechanical forces and depend on mechanotransduction for their daily survival. Piezo genes are not present in bacteria, but Piezo homologs can be found in protozoa, plants, and animals. Vertebrates have two Piezo genes—Piezo1 and Piezo2. Piezo genes in mammalians are expressed almost ubiquitously, suggesting a crucial role to mechanotransduction in different organ systems [38,39].

Evolutionary relationships of Piezo channels can be analyzed by phylogenetic tree construction using protein and genetic sequences from different species for understanding the relationships of Piezo channels [40]. The differentiation of Piezo genes was earlier than the evolution of species, which means that the genetic functions of Piezo1 and Piezo2 are different and unique [41]. The presence of more than one Piezo gene leads to potential additional functions of the Piezo gene which is no longer affected by evolutionary pressure. The observed structural differences between species could be explained with small optimizations for given circumstances in the different organisms. The core function of the Piezo is still the same.

## 4. Structure and Function of Piezo1

### 4.1. Channel Structure

The evolutionally conserved Piezo1 channel is a large monotrimeric transmembrane protein composed of approximately 2500 amino acid residues [39], with such a unique sequence that carries neither repetitive sequence pattern nor sequence homology to any other known cation channel families [42]. In humans, the PIEZO1 gene, also known as FAM38A/DHS/LMPH3/LMPHM6, is located on chromosome 16, containing 51 exons. This highly polymorphic gene has several variants [43]. In mice, where the vast majority of mammalian structure–function studies have been carried out, it is located in chromosome 8, containing 53 axons. These high resolution structural studies revealed that each subunit of the three-bladed, propeller-like protein made up of a peripheral blade containing 38 transmembrane (TM) helices, which are involved as mechanosensing module; a long beam on the intracellular side; and an anchor, which serves as a transduction module. Furthermore, a C-terminal domain (CTD) and a C-terminal extracellular domain (CED) form the ion-conducting pore module with the 37th (inner helix) and 38th (outer helix) transmembrane helices. The central cap is a trimeric complex formed by the CEDs of each subunit (Figure 2.). More detailed description can be found in [32,39,44,45].

### 4.2. Regulation

Considering the structural descriptions, a lever-like mechanism has been proposed to explain its exceptional mechanosensitivity, according to which the curved blades serve for mechanosensation, while the beam can act as a pivot coupling the extracellular blades to the central cap [46,47]. So, the force exerted on and the consequential conformation changes in distal blades is converted to the opening of the central pore that leads to nonselective cation permeation [48]. Piezo1 channels that mediate preferentially Ca2+ current, and are permeable for monovalent ions (K^+^, Na^+^ and Cs^+^), divalent ions (Ba^2+^, Ca^2+^, Mg^2+^ and Mn^2+^), and several organic cations (tetramethyl ammonium (TMA), tetraethyl ammonium (TEA)) [49,50].

The channels can be activated by pure mechanical stimuli, i.e., stretch, pressure, sheer stress, membrane torsion, bilayer indentation, osmotic stress [31], that can be modified by interactions with cytoskeletal proteins and linkages to the extracellular matrix elements [22,51]. The gating of the channel can be modeled with three states: open, closed, and inactivated [52]. The activation mechanism was illustrated by the membrane dome mechanism that was also experimentally proved [53,54]. The large extracellular blade domains of Piezo1 in its closed state deforms the neighboring cell membranes into a dome-like shape [47]. Upon mechanical stimulus, lateral membrane tension flattens the Piezo1 dome, which increases the energy of the membrane-channel system, resulting in the open state of the channel. This mechanism, however, does not take into account the current state and shape of the local membrane area. The footprint theory suggests that the Piezo1 channel induces deformity of the surrounding membrane which magnifies the sensitivity of Piezo1 to membrane stretch [55]. Piezo1 channels exhibit fast activation and voltage-dependent inactivation, and their kinetics show strong similarity in different cell types [56]. On the one hand, membrane stretch has been experimentally shown to be sufficient alone to activate mammalian Piezo1 proteins [57,58], without any further effect. On the other hand, several other interacting partners and cell factors have been identified that shift the activation threshold. The lipid composition of the membrane [59,60,61,62], as well as the alteration in substrate stiffness [63], in matrix roughness [64], environmental confinement [65], and the composition of the extracellular matrix [66] have all been shown to modify Piezo1 activity. Voltage sensitivity of Piezo1 was also demonstrated; its open probability is increased at positive voltages [67] due to the slow inactivation [1]. Acidic extracellular pH (~pH 6.3) was shown to diminish Piezo1 currents by stabilizing the inactivated state [52]. In addition to mechanical and environmental effects, protein–protein interactions have also been described as activating the Piezo1 channel. Stomatin-like protein 3 (STOML3) have been identified that can sensitize Piezo1 channels [68,69]. Cytoskeletal proteins with the potential to tune membrane rigidity, such as actin in several cell types, filamin A in smooth muscle, or dynamin in chondrocytes, were also proven to alter Piezo1 current [5,51,70]. Inhibition of actin polymerization by treatment with cytochalasin D also led to a decrease of Piezo1 current [71]. The effect of its interaction with the SERCA (endoplasmic reticulum Ca^2+^ pump sarco/ER Ca^2+^ ATPase) was also reported as modifying Piezo1 opening upon mechanical stimulation [72]. An inhibitory effect of Piezo1–Polycystin2 interaction on Piezo1 activity has also been suggested [73]. In turn, there is a substantial number of observations suggesting the feedback regulation of Piezo1 on cytoskeletal dynamics, therefore playing a role in cellular processes, such as cell migration [74]. The tethering of Piezo1 to the actin cytoskeleton via the cadherin-β-catenin mechanotransduction complex could explain both long-distance mechanogation of Piezo channels and the reverse effect as well, i.e., how Piezo1 proteins can be involved in cellular dynamical processes, such as cell proliferation, and migration [75].

### 4.3. Pharmacology

Although the main stimuli for Piezo1 channels are mechanical effects, a few pharmacological agents have also been proven to be able to modulate Piezo1 channel activity in the absence of mechanical stimulation. However, the list is not very long yet (Table 1).

#### 4.3.1. Agonists

A high throughput chemical library screening study identified Yoda1 (2-[5-[[(2,6-dichlorophenyl)methyl]thio]-1,3,4-thiadiazol-2-yl]-pyrazine), a small synthetic molecule as activator of Piezo1 channel (with no effect on Piezo2) [76]. Functional studies revealed that Yoda1 is a specific agonist of both mouse and human Piezo1 channels. Yoda1 modifies Piezo1 sensitivity and slows down the inactivation phase of the mechanically induced responses and partially activates the channels in the absence of external stimuli. Two additional, structurally different Piezo1-specific pharmacological agonists were identified: Jedi1 and Jedi2 [77], activating both human and mouse Piezo1. Co-administration of Yoda1 and Jedi1 resulted in synergistic activation, suggesting that they may act through different binding sites. Most likely, Yoda1 acts on the downstream beam, while Jedi1 on the upstream blade [77].

#### 4.3.2. Antagonists

Gadolinium and ruthenium red, as nonspecific blockers of mechanosensitive ion channels, have also been shown to block mouse Piezo1 channels [57,78].

GsMTx4, a tarantula venom peptide toxin, was found to inhibit cationic mechanically activated channels [50], including Piezo1 channels. GsMTx4 moderates the local membrane tension in the vicinity of the channel, thereby reducing the effective mechanical stimuli exerted on Piezo1 [79].

Dooku1, a Yoda1 analogue without agonistic effect, reversibly antagonizes Yoda1 activation [80]. Although, according to the original concept, Dooku1 has an inhibitory effect only on channels activated by Yoda1, several studies report that an antagonistic effect was also observed in the case of constitutively active channels [6,80,81]. It is worth mentioning how open Piezo1 channels can be considered constitutively active due to the influence of cells on each other in tissues compared to the measurements performed on individual, isolated cells.

A few fatty acids, rarely used in experiments aimed at the role of Piezo1 in cell physiological processes, were also found to be capable of non-specifically inhibiting the Piezo1 channel, such as margaric acid, arachidonic acid, and eicosapentaenoic acid [61].

A detailed summary on the regulation of Piezo1 channels can be found here: [39,45].

## 5. Piezo1 Channels in the Musculoskeletal System

The musculoskeletal system is one of the largest tissues and organ systems of the human body. The coordinated functioning of bones, joints, tendons, and muscles ensures posture and mobility. Limited functioning of any part leads to movement difficulties and pain, and in extreme cases, immobility or even death.

### 5.1. Skeletal Muscle

Skeletal muscle provides posture against gravity and the ability to move and breathe. It is a type of voluntary striated muscle, a very precisely organized tissue made up of myofibrils. A few myofibrils form a muscle fiber (myofiber), which is bounded by a sarcolemma and represents the functional unit, the muscle cell. These are multinucleated cells created by the fusion of myoblasts during development. Skeletal muscles are highly vascularized and innervated. They have an extraordinary regenerative capacity, which can be attributed to skeletal muscle stem cells (MuSCs) [82]. During regeneration, a part of the MuSCs (also known as satellite cells) is activated from a resting state. Activated MuSCs are characterized by the expression of PAX7, the canonical marker for satellite cells that coordinates the signalization of de novo myofiber formation and of myogenic regulatory factors (MYOD1, MYOG, and MYF5) during adult myogenesis [83]. These newly formed myoblasts (Pax7^+^/Myf5^+^/MyoD^+^) then either create new muscle fibers or fuse with existing ones promoting muscle regeneration [84,85], which is crucial for maintaining proper muscle function.

Since skeletal muscles are exposed to a series of mechanical stimuli because of their continuous contraction and relaxation, the role of mechanically activated channels in their physiological function naturally arises. It is therefore not surprising that even early studies highlighted the role of mechanically activated channels in skeletal muscle, based on measurements made on chick skeletal muscle [25]. A few years later, Franco and Lansman carried out experiments on mdx mice and confirmed the role of stretch-activated receptors in the development of muscular dystrophy [86], and then the effect of channel blockers on muscles from these animals was also examined [87]. At that time, the Piezo family had not yet been identified. The expression of Piezo1 in skeletal muscle was first demonstrated by Tsuchiya et al., on the C2C12 cell line [88]. Shortly thereafter, expression of the channel was also detected in the flexor digitorum brevis (FDB), tibialis anterior (TA), extensor digitorum longus (EDL), and soleus muscles [89,90,91,92]. All these studies point to the fundamental role of the Piezo1 channel in the mechanotransduction of skeletal muscles, thereby maintaining physiological function [93,94].

Nevertheless, investigations on the role of the Piezo1 channel in skeletal muscle relate primarily to its impact on myogenesis, muscle regeneration, and muscle development.

The continuous contractile activity of skeletal muscle fibers provides a constant direct mechanical stimulus for MuSCs. This mechanical effect is one of the essential stimuli for the activation of MuSCs. Activated MuSCs lead to myoblasts, a proliferating progenitor cell characterized by the expression of PAX7, MYOD, and MYF5. By the end of the terminal differentiation myocytes are formed, the fusion of which results in myotube formation. The role of the Piezo1 channel has been demonstrated in several steps of myogenesis. The expression of the channel can already be detected on the satellite cells before their activation; thus, importance of the channel can be assumed from the early phase of the development of satellite cells [90,91,95]. Indeed, according to the studies of Ma et al. and Hirano et al., which were performed in MuSC-specific Piezo1-deficient mouse strains, the knockout of the channel is associated with a significant decrease in the number of MuSCs, and the dynamics of activation in the remaining MuSC population was also significantly perturbed. The activation of the protein Rho-GTPase, which contributes to cytoskeletal rearrangement was hypothesized as the key molecule of signal transduction pathway of Piezo1. On the other hand, pharmacological activation of the channels by Yoda1 in the time window when most of the progenitor cells are still uncommitted, resulted in the same ratio of Pax7-positive cells as under control conditions, indicating no impact on MuSC proliferation. These data indicate that Piezo1 is required to maintain the MuSC pool. This is also supported by the study by Peng et al. [91] in which Piezo1 deficiency in TA muscle fibers caused a decline in the total number of Pax7^+^ cells and a relative increase in proliferating progenitor cells, which suggests that in the absence of Piezo1 activity, the cell is activated more rapidly. However, Peng et al. also reported that Yoda1 treatment reduced the number of myoblasts without affecting the total number of Pax7^+^ cells. The effect of the extracellular matrix stiffness on proliferation of MuSCs through the activation of Piezo1 was also demonstrated [95]. The satellite cells of Piezo1-deficient animals were not significantly affected by the stiffness of the culturing substrate, while the control cells showed increased proliferation with decreasing substrate elasticity.

Although proliferation was not clearly modified by Yoda1-induced activation of Piezo1, further steps of myogenesis were significantly altered. The increased fusion index of satellite cells in myocytes derived from freshly isolated FDB fibers following Yoda1 treatment indicates an accelerated rate of myotube formation [89]. The importance of Piezo1 in cell fusion was also confirmed by Piezo1 silencing: the fusion index responded with a significant decrease both in vivo and in vitro [91,92]. The significance of the Piezo1 channels in this phase may lie in the fact that during terminal differentiation and fusion, the cell membrane undergoes transformations that cause a constantly changing tension in it. These changes, considering any possible activation mechanism of the Piezo1 channel (force from filament or force from lipids), lead to a shift in the opening probability of the channel.

Modulation of Piezo1 channels can also influence the development of myoblasts through the modification of reactive oxygen species (ROS) production. It is known that a low level of intracellular ROS in skeletal muscle promotes the proliferation/differentiation of myoblasts [96] as it supports muscle regeneration through gene expression mediated by CREB (cAMP response element binding protein) and then ERK1/2 stimulation [97]. Activation of Piezo1 channels, on the other hand, can initiate inflammatory cascades, which, among other things, can lead to the development of pathological conditions due to the inflammatory effect of large ROS production [8]. Based on this study, inhibition of Piezo1 can reduce inflammation.

The effects related to the Piezo1 channel in the steps of muscle regeneration are becoming more and more recognized. At the same time, its potential role in adult muscle fibers can only be estimated based on a few studies, and it is far from being obvious. After the first measurements based on RT-PCR [88], the existence of Piezo1 clusters was described in isolated mouse FDB muscle fibers [89] and then in Gastrocnemius muscle fibers [98]. However, very little is known about their physiological or pathophysiological role. According to one of the most significant observations, the expression of Piezo1 is reduced in disused atrophied muscles [98]; at the same time, by analyzing human muscle biopsies of patients undergoing cast fixation after bone fracture, it was also revealed that atrophy-related genes are also upregulated. These data suggest a role for Piezo1 in muscle mass regulation. The possible role of the channels in adult muscle function is yet to be explored in detail.

### 5.2. Bone

Bone is a specialized, multirole and highly dynamic connective tissue which protects internal organs, supports the skeletal muscles system, and functions as a general scaffolding for structural and movement purposes [99]. Constant bone remodeling and bone mass maintenance is done by bone-resorbing osteoclasts, and the generation of new bone tissue by osteoblasts [100,101,102]. It has been previously described that bone actively responds to mechanical load. In bone mechanobiology there is the so called “Wolff’s law”, which states that bone grows and remodels in response to applied forces [103]. Endogen, exogen, biological, or gravitational mechanical stimuli have been widely recognized as vital elements for bone formation [104,105,106,107,108,109]. Recent results in skeletal research have showed that Piezo channels have significant effect on bone development and mechanical signaling. The expression of both Piezo1 and Piezo2 was detected in bone, but Piezo1 had higher expression than Piezo2 [48,110,111,112,113].

As global deletion of Piezo1 is embryonically lethal in mice [114], a skeletal tissue-specific Piezo deletion was necessary to investigate the effect of Piezo channels at different bone developmental stages. Piezo1Prx1-Cre mice (bone specific Piezo1 knockout) was used to investigate bone development and the bone loss was connected to increased bone resorption [113,115]. These results suggest that Piezo1 contributes minimally to skeletal patterning but is critical for bone formation during skeletal development, and it may be connected to abnormal mechanical loading [116]. Interestingly, mechanical unloading from tail suspension led to bone mass loss in control mice but not in Piezo1Prx1 mice.

A specialized stem cell, the Osteolectin+ cell in bone marrow, was reported to have a fundamental role in regulating bone mass. Piezo1 deletion in 2-month-old Osteolectin^+^ mice showed reduced bone mineral density and bone thickness in the cortical area [117].

Runt-related transcription factor 2 (Runx2) is a master regulator for the commitment of MSCs to osteoblastic differentiation. Piezo1 deletion in Runx2-expressing MSCs presented no changes in calvarial bone and had normal bone mass in the vertebral body at birth (postnatal day 0 or P0). The first rib fracture occurred at P5, and limb fractures followed at P14. Both male and female mice had shortening in the long-type bones and also had pelvic dysplasia. Large-scale reduction in trabecular bone mass was observed after P14. These types of bone development defects are associated with abnormal osteoblast functions, which led to an altered appearance and flattened osteoblasts covering the surface of trabecular bone. Significantly decreased serum P1NP (Procollagen 1 Intact N-Terminal Propeptide) levels and procollagen type I carboxyterminal propeptide levels were measured [111,118,119].

Osterix, a preosteoblast marker, is highly expressed in osteoblast and their progenitors. Osterix-positive cells with Piezo1 deletion caused rib fractures in 3-week-old mice without causing any observable bone defects at P0. These mice showed a reduced Osterix expression with decreased trabecular and cortical bone mass [113,115,120,121].

Collagen type 1 (Col1) is expressed throughout the pre-osteoblast stage to the differentiated osteoblasts. In 10-week-old mice, the deletion of Piezo1 in osteoblasts without Col1 (Cre/ERT) caused decreased trabecular bone mass, reduced cortical thickness, and insufficient collagen production. These mice also showed elevated rate in bone resorption [122].

Osteocalcin has high expression levels in mature osteoblasts [121]. Bone specific osteocalcin knock-out mice were generated and used to study Piezo1 deletion. These mice had non-altered skeletal sizes compared to their control littermates but had incomplete cranial suture closure. After 8 and 16 weeks, Piezo1Ocn mice showed significant bone mass loss and had shorter and weaker long bones. These changes resulted in smaller stature and decreased body weight. On the cellular level, osteoblasts had reduced differentiation in the bone of osteocalcin knock-out mice. Mechanical loading and unloading had no effect on these osteoblasts and osteoclasts [48]. These results support that Piezo1 in osteoblasts contributes to mechanosensation, and thus it is essential in proper bone development.

Dmp1 is a well-characterized marker which is expressed both by osteocytes and mature osteoblasts. Deletion of Piezo1 in Dmp1-Cre genetically modified mice resulted in severe osteopenia. These mice had normal body weights from birth to the age of 12 weeks. Dmp1-Cre mice with Piezo deletion had normal femur length, but the bone matrix was altered, which led to bone stiffness. Different research groups observed spontaneous fractures in the tibiae at 12 weeks of age, but others reported no such spontaneous fractures in these types of mice. This could be explained by the different Dmp1 promoters used [123,124]. The above-mentioned changes could be explained by decreased bone formation efficacy or increased bone resorption rate [112]. Based on these findings, Piezo1 in osteocytes has a significant effect on mechanotransduction under different load conditions as it could activate bone formation or inhibit bone resorption.

In the developmental process of bone dysregulation in bone building and destroying components can lead to developmental disorders. Piezo1 expression in osteoclasts can also affect skeletal function. Cathepsin K-Cre Piezo1-depleted mice presented with no difference in stature and body weight, had normal bone mass, and were unaffected in the bone resorption process [115]. These results could be explained by the low Piezo1 expression characteristic for the osteoclast-like cell lines.

Double MSC-specific KO for Piezo1 and Piezo2 (dKO) were generated and showed more severe skeletal defects compared to Piezo1 single-KO. dKO mice were diagnosed with shorter long bones and more severe bone loss in cortical and trabecular bone, reduced bone formation rate, decreased Osterix expression, and P1NP serum level [113]. However, deletion of Piezo2 in osteoblasts had no adverse effect in Prx1-Cre or Osterix-Cre mice. These mice showed close to normal skeletal development at P0 and no detectable length difference in long bones and had normal bone mass at later developmental stages. Piezo2 deletion caused no fractures in mice [112,113]. Low expression of Piezo2 could explain that in osteoblasts and osteocytes Piezo2 alone has very limited contribution to bone homeostasis [110]. The above described findings suggest a functional redundancy and exchangeability of Piezo1 and Piezo2 in bone homeostasis.

Piezo1 activation could lead to beneficial consequences. Two-week-long Yoda1 treatment on 4-month-old female C57BL/6J mice with 5 μmol × (kg body weight) − 1 increased bone mass without significantly altering body weight or osteoclast bone resorption. Low-intensity pulsed ultrasound (LIPUS) can stimulate Piezo1 channel activation in osteoblastic cell lines [125]. LIPUS can lead to Piezo1 channel opening and elevated calcium influx and activate the extracellular signal–regulated kinase 1/2 (ERK1/2) and F-actin polymerization. These molecular changes helped cell migration and increased proliferation [126].

### 5.3. Tendons and Ligaments

Tendons connect muscle to bone and are essential for musculoskeletal function, including athletic performance. The role of mechanical stress transmitted by or to tendons in physical performance is not fully clarified. Nevertheless, mechanosensitive channels may also play a significant role in tendons and ligaments.

Nakamichi et al. generated tendon-specific knock-in mice using R2482H Piezo1, a mouse gain-of-function variant, and found that they had higher jumping abilities and faster running speeds than wild-type or muscle-specific knock-in mice. Tendon anabolism had been increased by tendon-specific transcription factor elevation, but there were no similar changes in skeletal muscle. Piezo1 facilitates tendon tissue formation and enhances tendon enlargement. Biomechanical investigations showed that the tendons of R2482H Piezo1 mice were more compliant and stored more elastic energy, consistent with the increased jumping ability. These phenotypes were replicated in mice with tendon-specific R2482H Piezo1 replacement after tendon maturation, indicating that Piezo1 could be a promising target for improving physical performance by enhancing function in mature tendons. The frequency of E756del Piezo1 was higher in sprinters than in population-matched nonathletic controls in a small Jamaican cohort, proposing similar functions in humans. The results described a critical function of tendons in physical performance, which is strictly regulated by Piezo1 in tenocytes [127]. The group and the Athlome Consortium researchers collected genetic information on elite athletes. In the Athlome database, 46% had one copy of Piezo1 with the tendon-impacting mutation and 8% had two copies of the mutation among 91 Jamaican sprinters. Of Jamaican students who had not competed in track events, 31% had one version of the mutation and 2% had two copies. Sprinters having two copies of the mutation showed similar trends among Greek athletes, with 3 to 5 times more than controls, and 1.3 to 1.75 times more compared sprinters having only one copy of the mutation. The researchers suggest that targeting the protein could help in the treatment of tendon injuries or reduce age-related declines in mobility [127].

Passini et al. investigated tendon mechanotransduction by combining Ca^2+^ imaging with simultaneous mechanical loading of tendon explants and isolated primary tendon cells [128]. In their models shear stress triggers the Ca^2+^ influx in isolated tenocytes. Cells with depletion of Piezo1 showed a significant reduction in the shear-stress response from all examined knockdowns. Piezo1 knock-downs in rat tenocytes isolated from tail tendon fascicles showed similar results, suggesting the importance of Piezo1 as shear-stress sensor in tenocytes. Tendon stiffness is regulated by Piezo1-mediated mechanosignaling in vitro and in vivo as well, most likely via denser collagen cross-linking. In addition to that, Piezo1 activity influences jumping performance in humans. Actually, about one out of three individuals of African descent carries a PIEZO1 gain-of-function (GOF) mutation known as E756del. This mutation is associated with malaria resistance and represents the most abundant PIEZO1GOF mutation identified to date. Normalized jumping performance of patients with mutations showed a net 13.2% average increase compared to non-carriers [128].

Tendons also play an important role in proprioception. The perception of body and limb position is mediated by proprioceptors, specialized mechanosensory neurons that transmit information about the stretch and tension experienced by muscles, tendons, skin and joints. There are several studies which found that Piezo2 was expressed in sensory endings of proprioceptors innervating muscle spindles and Golgi tendon organs in mice. The results indicate that Piezo2 is the major mechanotransducer of mammalian proprioceptors [114,129,130].

No data are available on Piezo1 and Piezo2 in ligaments of the synovial joints. However, there are studies which evaluate the function of Piezo1 in periodontal ligament (PDL) tissue, which lies between tooth cementum and alveolar bone and plays a fundamental role in bone homeostasis. Forces affecting these structures can lead to surrounding bone remodeling. In the study by Jin et al., primary human PDL cells (hPDLCs) were isolated, cultured, and then subjected to static compressive loading [131]. They observed that Piezo1 mRNA increased after 0.5 h of mechanical loading and lasted for 12 h. Imunofluorescence and Western blot analysis confirmed these changes after stimulation. Results indicate that Piezo1 ion channels contribute to transduction of the mechanical stress, inducing osteoclastogenesis.

Other studies have confirmed that the Piezo1 ion channel can transmit mechanical signals and regulate both the osteogenic differentiation and osteoclastogenesis of PDL. Several potential signaling pathways have been proposed, including extracellular regulated protein kinases (ERK), nuclear factor-kappa B (NF-κB), and Notch1 signaling pathways [132,133]. These results suggest a possible role of Piezo1 in orthodontic tooth movement.

### 5.4. Joints

The joints are complex structures connecting adjacent bones. In the human body they are categorized into three groups: synarthroses (immovable), e.g., the skull; amphiarthroses/cartilaginous (slightly movable), e.g., the vertebrae; diarthroses/synovial (freely movable), e.g., the knee.

The roles of synovial joints in weight bearing and locomotion are essential. The joint cartilage is exposed to millions of mechanical cycles of different levels of strain. Chondrocytes, while accommodating these effects, must regulate their metabolic activities according to mechanical loading. Chondrocytes are responsible for maintaining and remodeling cartilage. Pathological mechanical stress can lead to maladaptive responses and subsequent cartilage degeneration.

#### Piezo1 in Chondrocytes

Studies have identified TRPV4 and Piezo channels to function synergistically in chondrocyte mechanotransduction in response to injurious mechanical loading [70,134]. The function of the Piezo channel as the major regulator in mechanotransduction is to regulate calcium signaling and maintain the cartilage matrix. An atomic force microscopy study demonstrated the contribution of Piezo1 and Piezo2 in calcium signals in compressed porcine chondrocytes [135]. Apoptosis of chondrocytes acts as a critical mechanism leading to subsequent post-traumatic arthritis following joint injuries causing trauma. Piezo1 has been proposed as a potential molecular target for reducing cell death and injury-induced cartilage degeneration. However, to prove these theories, further investigations are required.

Lee et al. confirmed significant Piezo1/2 expression levels in both porcine and human primary chondrocytes. They also reported Ca^2+^ influx into chondrocytes via Piezo channels evoked by mechanical load, and the inhibition of Piezo with the peptide GsMTx4 protects articular chondrocytes from mechanically triggered cell death. It was also suggested that vulnerability of chondrocytes to mechanical trauma might be related to Piezo1/2-mechanosensitive ion channels [135]. They found significantly increased Piezo1 messenger RNA (mRNA) in porcine primary articular chondrocytes in response to pathologically relevant IL-1α concentrations and elevated Piezo1 protein expression. In human osteoarthritic cartilage Piezo1 expression has been detected to be significantly elevated compared to healthy human controls. Piezo1 function was also tested using Yoda1, and increased Piezo1 function was observed by Ca^2+^-imaging on chondrocytes exposed to IL-1α [41]. Low levels of IL-1α, even at subnanomolar concentrations that correspond to early stages of osteoarthritis (OA) pathogenesis, make chondrocytes more sensitive to dynamic mechanical compression. Overexpression of Piezo1 channels causes elevated resting [Ca^2+^]i and mechanical-stress-induced Ca^2+^ influx, indicating signs of mechanical hypersensitivity, “hypermechanotransduction”. Increased resting [Ca^2+^]i, via Piezo1, leads to decreased F-actin density and decreased stiffness of the chondrocytes. This signaling results in intense cellular deformation in response to mechanical loading. This positive feedback loop could play a role in the initiation and progression of OA [135].

It is also well known from clinical practice that hyperphysiological overload and injurious loading of the joints can lead to OA. Abnormal mechanical loading increased the expression of Piezo1 in cultured human chondrocytes showing excessive Ca^2+^ influx. The cytoplasmic Ca^2+^ activated endoplasmic reticulum stress upregulated the expression of caspase-12 and subsequent chondrocyte apoptosis [136]. Inhibition of Piezo1 by GsMTx4 prevented high strain-induced cell death. Urocortin1, a corticotropin-releasing factor-related peptide found in chondrocytes was identified as chondroprotective agent preventing the Ca^2+^ overload by keeping Piezo1 in a closed state [137].

Piezo1 and Piezo2 proteins were observed in mouse primary chondrocytes by Du et al. [138]. In contrast, Servin-Vences et al. reported that only Piezo1 expression could be reliably measured, and Piezo2 was not detectable. They found that both TRPV4 and PIEZO1 participate in mediating mechanically activated currents in chondrocytes [27].

Findings about Piezo-mediated mechanotransduction and mechanosensation in chondrocytes could be one of the most important steps in developing therapeutic solutions to OA.

### 5.5. Intervertebral Discs

The intervertebral discs lie between the vertebral bodies, linking them together. The components of the disc are the nucleus pulpous, annulus fibrous, and cartilaginous end-plates. The blood supply to the disc is limited only to the cartilaginous endplates [139]. As the disc ages, degeneration occurs [139]. The relationship between aging, degenerative processes, and actual illness is not fully understood.

In the degenerated vertebral disc tissue or in response to mechanical stress Piezo1 overexpression has been detected both at mRNA and protein level in the cells of the nucleus pulpous [140,141,142,143,144,145]. Activation of Piezo1 channels has been shown to convert mechanical stress into inflammatory signals [141,142,145]. Inflammasome NLRP3 (Nod-like receptor family pyrin domain containing 3 protein) is activated by Piezo1 signaling, leading to the production of interleukin-1β (IL-1β) [141,142]. Suppressing Piezo1 activity by siRNA technique led to reduced proinflammatory activities and less abnormal metabolism [142].

Excessive mechanical load in parallel with an inflammation can induce apoptosis in cells of the nucleus pulpous [146]. This phenomenon is similar to that in chondrocytes; however, the exact underlying mechanisms are still unknown. Apoptosis was assigned to Piezo1 activation [140,145] leading to alternating mitochondrial membrane potential. Mechanical stretch applied to nucleus pulpous cells in a monolayer resulted in changes of mitochondrial potential and increased apoptotic rate, while suppression of Piezo1 via short hairpin RNA (shRNA) transfection lessened these changes. Mitochondrial dysfunction and increased apoptosis has also been shown after human intervertebral disc compression, partially rescued by treatment with GsMTx4 [145].

Intervertebral disc (IVD) cells and extracellular matrix (ECM) are associated to form a well-organized functional system. ECM components are produced mainly by IVD cells, while IVD cell phenotype and viability are modulated by ECM through direct interactions and indirect regulations [139].

In human degenerated nucleus pulpous samples, increased ECM stiffness can be observed. Furthermore, in degenerated disc samples, concurrent overexpression of Piezo1 were also present. Accumulation of ROS has been observed in the background of increased substrate stiffness, and this accumulation led to apoptosis, autophagy, and senescence [143,144,145]. In turn, knocking down Piezo1 in the nucleus pulpous cells reduced senescence, oxidative stress, and endoplasmic reticulum stress to soft substrate levels, while Piezo1 activation increased the production of periostin [143,144,145], which triggered senescence in stiff, degenerated NP tissue. Periostin has a crucial role in activating a positive feedback loop in degeneration of intervertebral discs. Periostin activates the nuclear factor kappa B pathway, which leads to an elevated periostin expression. This loop can be triggered by Yoda1 and blocked by GsMTx4 [144].

Piezo1 signaling has been linked to activation of every intracellular degenerative processes such as inflammation, apoptosis, and senescence. Therefore, Piezo1 may be a promising therapeutic target in the treatment of low back pain caused by degeneration and senescence.

## 6. Disorders in the Musculoskeletal System Associated with Piezo1 Channels

As mentioned in the introduction in connection with several organ systems and discussed in detail regarding the musculoskeletal system, the Piezo1 channels fundamentally contribute to the healthy function of the human body. Several human diseases have been associated to mutations, malfunctions, or expression deficits of the channel. In Table 2 known Piezo1-related diseases are listed by affected organs.

## 7. Conclusions

Evidence relating to the role of Piezo1 channels activated by mechanical stimuli in the musculoskeletal system is continuously increasing. It is now generally accepted that the regenerative capacity of skeletal muscles—although the details are not yet fully known—is significantly dependent on the functioning of the Piezo1 channels. It also turned out that the Piezo1 channel contributes to the regulation of muscle trophism. We have known for a long time that physical exercise helps to maintain muscle functions, but it has now been discovered that, at least in part, this positive effect is established through the activation of Piezo1 channels. Furthermore, the importance of Piezo1 channels was revealed not only in the muscles but also in the bones, tendons, and cartilage.

The operation of the Piezo1 channels is also related to the properties of the cell membrane, so its state, which is modified pharmacologically or as a result of nutrition, also indirectly affects muscle functions. Factors released during inflammatory processes can also affect the channel, thereby affecting the condition of the muscle, intervertebral disk, or cartilage. Based on these observations, Piezo1 channels can be potential therapeutic targets for disorders related to muscle regeneration and atrophy and to degenerative processes of cartilage or IV disks. The discovery of additional potential regulatory mechanisms between the Piezo1 channel and musculoskeletal system function may further broaden the field of application.

However, tissue-specific targeting is essential for the development of effective therapy due to the ubiquitous expression of Piezo1 channels. Since the activation of the channel in certain organs has benefits, in other tissues a positive effect can be achieved by inhibiting it. Furthermore, the stimulation and inhibition of the channel within the same organ can have a positive result, so to achieve a beneficial combined outcome, coordination of several circumstances and much additional research are necessary. Nevertheless, the Piezo1 channel is a promising target for the development of new types of drugs in musculoskeletal disorders. In our opinion, the role of mechanosensitive adaptation and mechanically activated channels in the functioning of the locomotor system, especially in muscle function, may be significantly more prominent than currently known. Piezo1, as a calcium channel, might be a currently poorly understood regulator of calcium homeostasis in these cells.

## Figures and Tables

**Figure 1 ijms-24-06513-f001:**
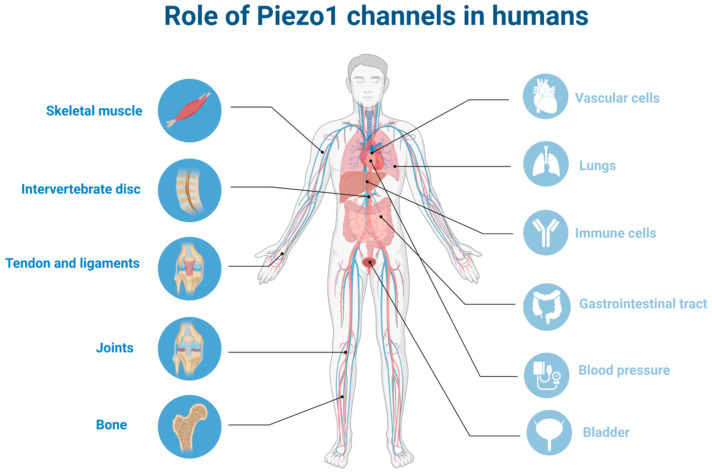
Role of Piezo1 channels in humans: localization of Piezo1 in different organs focusing on musculoskeletal system (the figure was created using biorender.com, accessed on 23 March 2023.).

**Figure 2 ijms-24-06513-f002:**
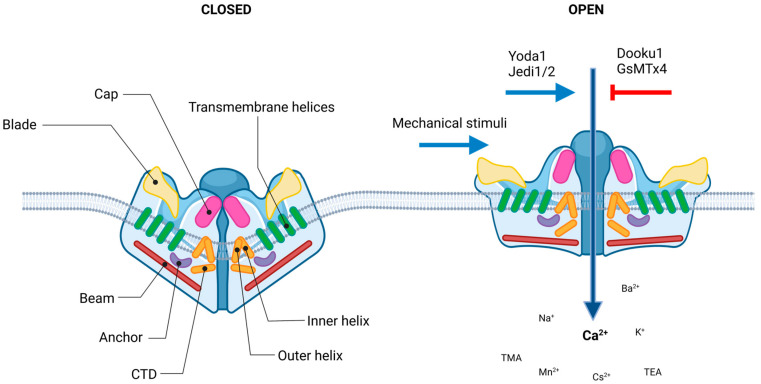
Piezo1 channel. Structure and regulation of the channel. Piezo1 protein has a homotrimeric architecture with three propeller-shaped blades. The pore consists of the extracellular Cap structure, three pairs of the outer and inner helices (TM domains), and the intracellular C-terminal domain (CTD). The peripheral mechanotransduction region contains the beam structure, the peripheral blade, and the anchor domain (the figure was created using biorender.com).

**Table 1 ijms-24-06513-t001:** Pharmacological modulators of Piezo1 channel. Bold letters indicate Piezo1-specific agents.

Agonists	Antagonists
**Yoda1**	Gadolinium
**Jedi1, Jedi2**	Ruthenium red
	GsMTx4
	**Dooku1**
	Margaric acid
	Arachidonic acid
	Eicosapentaenoic acid

**Table 2 ijms-24-06513-t002:** Piezo1 channel associated diseases.

Organ System	Affected Organs	Diseases	Affected Cell Type	Experimental Setup	References
Skeletal system	Bone	Osteoporosis	Osteocytes, osteoblasts	In vitro Human and mouse MSCs	[147]
	Bone	Osteosarcoma	Human osteosarcoma cells	In vitro Human osteosarcoma cell line MG63 and U2 cell line	[148]
	Joints	Synovial sarcoma	SW982 cells	In vitro Human synovial sarcoma SW982, human embryonic kidney 293 cell lines	[149]
	Joints	Joint disease	Chondrocytes	In vitro Primary chondrocytes from six-day-old mice	[138]
	Dental system	Orthodontic tooth movement	Periodontal ligament cells	Sprague-Dawley rats (8 week old)	[131,150]
	Heart	Cardiovascular disease	Cardiac fibroblasts, cardiomyocytes	In vitro Knockout Mice sm22Cre Piezo1^−/−^ Mice	[151]
Gastrointestinal system	Gastic region	Gastric cancer	Gastric cancer cell lines	In vitro Gastric cancer cell lines SGC-7901 and BGC-823	[152]
	Pancreas	Pancreatitis	Pancreatic acinar cells	In vitro C57BL/6J male mice 6–8 week old and Piezo1 knockout mice: Ptf1atm2(cre/ESR1)Cvw/J mice	[153]
	Pancreas	Pancreatic ductal adenocarcinoma	Pancreatic stellate cells	In vitro Human pancreatic cancer cell lines PCs, MiaPaCa-2 and Panc-1	[154]
Excretory system	Bladder	Bladder dysfunction	Bladder urothelial cells	In vitro C57BL/6Cr) mice and TRPV4-knockout mice In vitro Patients with prostate cancer or benign prostatic hyperplasia	[155]
	Kidney	Renal fibrosis	HK2 cells	In vitro C57BL/6J mice In vitro Human kidney autopsy specimens Human proximal tubular cells (HK2 cells)	[156]
Respiratory system	Lungs	Acute respiratory distress syndrome	Type II pneumocytes	In vitro Health adult Sprague-Dawley rats	[157]
	Lungs	Lung capillary stress failure injury	Lung endothelial cells	In vitro Backcrossing Piezo1flox/flox mice with Endo-SCL-Cre mice	[158]
	Lungs	Lung cancer	Small cell lung cancer lines	In vitro 16HBE cells	[159]
Nervous system	Central nervous system	Gliomas	Human gliomas cells	Human glioma tissues	[160,161]
	Astrocytes	Neuroinflammation	Astrocytes	In vitro C57BL/6 mice; mixed glial cell cultures	[162]
	Eye	Glaucoma	Cornea, retinal ganglion cell layer, and lens epithelial cells	In vitro 7-week-old male albino ddY mice, 15-month-old C57BL/6 mice, 9- and 15-month-old DBA/2J mice, adult Sprague–Dawley rats	[163]
	Neurons	Migraine	Trigeminal ganglion	In vitro Wistar rats	[164,165]
Connective tissue	Skin	Hypertrophic scar	Myofibroblasts	In vitro Human hypertrophic scar tissues and adjacent normal skin tissues from 9 people	[166]
	Skin	Skin wound	Keratinocytes	In vitro Krt14Cre;Piezo1fl/fl (Piezo1-cKO) Krt14Cre;Piezo1cx/+ and Krt14Cre;Piezo1cx/cx (Piezo1-GoF)	[167]
	Adipose tissue	Inflammation	Adipocytes	In vitro Piezo1-flox/flox mice were crossed to adiponectin-Cre mice	[168]

## Data Availability

Not applicable.

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
