# Peer review of "The Role of the Piezo1 Mechanosensitive Channel in the Musculoskeletal System"

_ijms, 2023, doi:10.3390/ijms24076513_

Round 1
Reviewer 1 Report
The article presents comprehensive overview of Piezo1 mechanosensitive channel in the musculoskeletal system. Overall, the manuscript is well-written, I have only a few comments.
Specific comments are elucidated below:
1. Line 138, I think instead of Figure 1, there should be figure 2.
2. Regulation and the effect of pharmacologic agents could be summarized in a figure or table for easier overview.
3. SInce Gd3+ is a blocker of Piezo1 channels, is there any reports that Gadolinium containing MRI contrast agent could inhibit Piezo1 in vivo?
4. The introductory paragraphs of each sections are generally to general and not necessary, making the paper long and not focused. I suggest making the text a bit more concise to increase focus and clarity of the review.
5. Is there any data on effect of physical activity/age/obesity/diabetes/sex on Piezo1 expression in musculoskeletal tissues?
6. What about Piezo1expression and role in fascia?
7. There are no references from lines 493-507?
8. In table 1 please add whether the studies were performed in vitro/in vivo/in situ/in animal or human cells.
9. In conclusion section please state what in your opinion is the way forward in the knowledge of piezo1 expression/function/modulation in musculoskeletal system.
Author Response
Thank you for your critical comments and questions. We answer them one by one below.
Specific comments are elucidated below:
- Line 138, I think instead of Figure 1, there should be figure 2.
Thank you for your comment, we have corrected it.
- Regulation and the effect of pharmacologic agents could be summarized in a figure or table for easier overview.
According to your suggestion, we have inserted a small table
- SInce Gd3+ is a blocker of Piezo1 channels, is there any reports that Gadolinium containing MRI contrast agent could inhibit Piezo1 in vivo?
According to our information, there is no publication that examines the relationship between the gadolinium used in the medical examinations as contrast agent and the Piezo1 channel.
We assume that since the dose used in medical examinations is high, we cannot rule out that the effect on the Piezo1 channel may be responsible for the development of some side effects (joint pain, muscle weakness). At the same time, since gadolinium is a general calcium channel blocker, its direct effects on other calcium channels providing critical life functions (e.g. the L-type calcium channel of the heart) would have more serious consequences. Since such an effect is not known, it can be assumed that the modulation of the Piezo1 channels can be small. According to a recently published review on the gadolinium-based contrast agents (GBCAs) (Do, Q.N.; Lenkinski, R.E.; Tircso, G.; Kovacs, Z. How the Chemical Properties of GBCAs Influence Their Safety Profiles In Vivo. Molecules 2022, 27, 58. https://doi.org/10.3390/ molecules27010058): “At the present time, there are no rigorous studies that have shown association of gadolinium deposition with clinical symptoms or data that suggest that it is harmful to patients. Self-reported clinical symptoms of “gadolinium deposition disease” such as generalized sensory symptoms lack clinical evidence to exclude alternative causes for these symptoms. Published studies [154–156] suffered from considerable selection bias and a definite discordance between radiological evidence and individual clinical symptoms.”
- The introductory paragraphs of each sections are generally to general and not necessary, making the paper long and not focused. I suggest making the text a bit more concise to increase focus and clarity of the review.
The introductory parts of the paragraphs about the individual organs have been shortened.
- Is there any data on effect of physical activity/age/obesity/diabetes/sex on Piezo1 expression in musculoskeletal tissues?
To the best of our knowledge, there is no data on the direct dependence of the expression of the Piezo1 channel in the musculoskeletal system on age/fitness/diabetes/gender/obesity. The publication mentioned in the review is the only one we know of that shows a correlation between inactivity and the expression of Piezo1 in the muscle (Hirata, Y.; Nomura, K.; Kato, D.; Tachibana, Y.; Niikura, T.; Uchiyama, K.; Hosooka, T.; Fukui, T.; Oe, K.; Kuroda, R.; et al. A Piezo1/KLF15/IL-6 axis mediates immobilizationinduced muscle atrophy. J. Clin. Invest. 2022, 132, doi:10.1172/JCI154611.). However, since Piezo1 has been associated with senescence in many cell types, it probably has a role in aging, although the exact process is not yet clear.The current experiments of our working group attempt to shed light on such an aspect.
- What about Piezo1expression and role in fascia?
The term fascia is used in the literature to refer to the broad concept of connective tissue. In this sense there is data available (see red blood cells). On the other hand, no such data is available in the musculoskeletal system, where fascia means sheets of connective tissue that form interconnecting planes spanning the entire body, surrounding and separating muscles, and creating biomechanical interfaces between them. As stated in a recently publisehd paper of Langevin (Langevin HM. Fascia Mobility, Proprioception, and Myofascial Pain. Life (Basel). 2021 Jul 8;11(7):668. doi: 10.3390/life11070668. PMID: 34357040; PMCID: PMC8304470.):
„Piezo channels are mechanosensory ion channels that are important for joint proprioception [13] although their presence/role in fascia is currently unknown. Piezo2 deficiency syndrome (a form of arthrogryposis) is characterized by both joint hypermobility and contractures as well as muscle weakness. Patients with Piezo gain- or loss-of-function mutations have not been systematically investigated for connective tissue biomechanical properties although there are reports of joint laxity in individual patients and families [56,57].”
- There are no references from lines 493-507?
A significant part of the mentioned lines was deleted in accordance with the reviewer's former suggestion, in the text that remained the missing references were included.
- In table 1 please add whether the studies were performed in vitro/in vivo/in situ/in animal or human cells.
We have supplemented the table with the suggested information.
- In conclusion section please state what in your opinion is the way forward in the knowledge of piezo1 expression/function/modulation in musculoskeletal system.
In the final part, our personal opinion is now also formulated.
Reviewer 2 Report
The literature review on the Piezo channel provides a comprehensive overview of the current state of research in this field. The authors have done an excellent job of summarizing and analyzing a wide range of studies, highlighting key findings, and identifying gaps in knowledge. The review begins by introducing the Piezo channel and its importance in cellular mechanotransduction, providing a clear rationale for its study. Also the review gives some additional information about mechanosentive ion channels and their relationship with Piezo. The authors then proceed to explore the structural and functional characteristics of the channel, highlighting recent advances in the field.
One of the strengths of this review is its focus on the physiological relevance of the Piezo channel. The authors provide a detailed discussion of the role of the channel in various cellular processes, including vascular development, touch sensation, and bladder function. They also highlight the clinical implications of Piezo channel dysfunction, such as the development of hypertension and chronic pain.
In addition to summarizing the current literature, the authors also identify several areas where further research is needed. For example, they highlight the need for more studies on the regulation of the Piezo channel, as well as investigations into the relationship between Piezo channel activity and disease. However, the quality of figures could really improve. Second is the organization could improve in some aspect. I think the paragraph from line 110 discussed more details about the structure of Piezo channel rather than its role in animal world. These organization could be improved in multiple aspects.
Overall, this literature review is a valuable resource for researchers and clinicians interested in the Piezo channel and its role in cellular physiology and disease.
Author Response
Thank you for the work of the reviewer. We will respond to the comments below.
The literature review on the Piezo channel provides a comprehensive overview of the current state of research in this field. The authors have done an excellent job of summarizing and analyzing a wide range of studies, highlighting key findings, and identifying gaps in knowledge. The review begins by introducing the Piezo channel and its importance in cellular mechanotransduction, providing a clear rationale for its study. Also the review gives some additional information about mechanosentive ion channels and their relationship with Piezo. The authors then proceed to explore the structural and functional characteristics of the channel, highlighting recent advances in the field.
One of the strengths of this review is its focus on the physiological relevance of the Piezo channel. The authors provide a detailed discussion of the role of the channel in various cellular processes, including vascular development, touch sensation, and bladder function. They also highlight the clinical implications of Piezo channel dysfunction, such as the development of hypertension and chronic pain.
In addition to summarizing the current literature, the authors also identify several areas where further research is needed. For example, they highlight the need for more studies on the regulation of the Piezo channel, as well as investigations into the relationship between Piezo channel activity and disease. However, the quality of figures could really improve.
The images of the manuscript submitted for revision were taken in low resolution. We have improved the quality of the images.
Second is the organization could improve in some aspect. I think the paragraph from line 110 discussed more details about the structure of Piezo channel rather than its role in animal world. These organization could be improved in multiple aspects.
The text has undergone minor modifications. The repeated parts of the mentioned section (paragraph of line 110) concerning the structure of the channel have been removed. The introductions of the paragraphs relating to individual organs have been shortened.
Overall, this literature review is a valuable resource for researchers and clinicians interested in the Piezo channel and its role in cellular physiology and disease.
Round 2
Reviewer 1 Report
The authors satisfactorily addressed all my comments.